# ODCA-YOLO: An Omni-Dynamic Convolution Coordinate Attention-Based YOLO for Wood Defect Detection

**Rijun Wang [1,2]**, **Fulong Liang [1,2]**, **Bo Wang [2,3,\*]** and **Xiangwei Mou [1,2,\*]**

1   School of Teachers College for Vocational and Technical Education, Guangxi Normal University, Guilin 541004, China; wangrijun1982@126.com (R.W.); fl15350630563@163.com (F.L.)
2   Key Laboratory of AI and Information Processing, Hechi University, Yizhou 546300, China
3   School of Artificial Intelligence and Smart Manufacturing, Hechi University, Yizhou 546300, China
*   Correspondence: 05041@hcnu.edu.cn (B.W.); xwmou@mailbox.gxnu.edu.cn (X.M.)

**Abstract:** Accurate detection of wood defects plays a crucial role in optimizing wood utilization, minimizing corporate expenses, and safeguarding precious forest resources. To achieve precise identification of surface defects in wood, we present a novel approach called the Omni-dynamic convolution coordinate attention-based YOLO (ODCA-YOLO) model. This model incorporates an Omni-dimensional dynamic convolution-based coordinate attention (ODCA) mechanism, which significantly enhances its ability to detect small target defects and boosts its expressiveness. Furthermore, to reinforce the feature extraction and fusion capabilities of the ODCA-YOLO network, we introduce a highly efficient features extraction network block known as S-HorBlock. By integrating HorBlock into the ShuffleNet network, this design optimizes the overall performance. Our proposed ODCA-YOLO model was rigorously evaluated using an optimized wood surface defect dataset through ablation and comparison experiments. The results demonstrate the effectiveness of our approach, achieving an impressive 78.5% in the mean average precision (mAP) metric and showing a remarkable 9% improvement in mAP compared to the original algorithm. Our proposed model can satisfy the need for accurate detection of wood surface defects.

**Keywords:** defect detection; deep learning; wood defects; YOLOv7; attention mechanism





## 1. Introduction

Wood is a significant natural resource with wide-ranging applications in sectors including housing construction, furniture manufacturing, and the production of everyday essentials, among others. However, during the growth of trees, some unpredictable defects will be produced, as well as in the subsequent processing process, due to improper processing technology, which will also lead to some defects. Common defects in wood panels include dead knots, live knots, cracks, etc., and these defects greatly reduce the utilization rate of wood. Simultaneously, the wood structure's uniformity and integrity face disruption, leading to compromised surface aesthetics and processing performance. Furthermore, the strength of the wood is compromised, resulting in a reduced service life [1]. Utilizing wood with these defects in constructions or furniture can pose significant safety risks, including instability or potential breakage [2]. Hence, the identification of wood defects becomes indispensable for assessing the wood's usability before proceeding with further processing.

Wood defect detection techniques can be broadly categorized into two groups: those targeting internal wood defects and those focusing on surface defects. For detecting internal wood defects, various methods have been employed, including vibration detection [2,3], air-coupled ultrasound-based techniques [4,5], stress wave techniques [6,7], and X-ray techniques [8,9]. Vibration detection methods rely on the impact of wood's internal structure on vibration frequency to assess the presence and severity of defects. However, accurately identifying the specific type and location of defects remains challenging. On the other hand,

air-coupled ultrasound-based techniques leverage the impact of internal defects on ultrasound propagation speed to assess the condition of defects within the wood. Since the ultrasound is easily affected by the external environment and the internal grain of the wood, the detection accuracy and efficiency are reduced. The stress wave techniques propagate rays through the stress wave segments to reconstruct images of internal defects. Nonetheless, implementing this approach in industrial production processes is not practical due to the need for multiple sensors to be affixed to the wood surface for measurements, which hampers efficiency in the detection process. The X-ray techniques use the fact that defects within the wood will absorb different rays than normal parts of the wood, in order to detect defects. X-ray techniques, besides being expensive, also pose a risk of radiation exposure to the operator, thereby restricting their practical usage. In brief, the methods used to detect internal wood defects typically assess the severity of these defects but do not provide precise identification of the specific defect types present in the wood.

Surface defect detection in wood includes various methods, including manual visual recognition [10], surface defect detection utilizing 3D laser scanning technology, and techniques based on image recognition [11,12]. In the traditional wood processing approaches of the past, inspectors primarily relied on manual visual recognition to identify surface defects on wood. This identification method is easy to produce visual fatigue, resulting in leakage and false detection. In addition, the detection efficiency of this identification method is low. The principle of surface defect detection based on 3D laser scanning technology is that the laser source emits laser light into the wood, which is reflected by the wood and then received by the laser detector. Computer software analyzes and processes the received laser rays to obtain detailed surface data of the wood, enabling the classification of defect types. This detection method takes advantage of the irregularity of the defective surface. It is better at detecting defects such as wormholes, cracks, and other depressed or raised structures of the surface [13]. However, this method is less satisfactory in detecting defects with subtle changes in the three-dimensional surface structure, such as live knots or discolorations. In addition, this method often requires manufacturers to purchase more expensive laser inspection equipment. The surface defect detection method based on image recognition operates on a simple principle. A camera captures an image of the wood surface, which is then analyzed by an image recognition algorithm in a computer to detect and identify defects. The advantage of this approach lies in its simplicity, as it does not necessitate the use of additional equipment, resulting in cost savings. Moreover, it is largely operator-neutral, making it a highly preferred choice for practical wood processing applications [14].

In recent years, deep learning has gained significant traction in target detection, especially with the advancements in convolutional neural networks (CNNs). Consequently, deep learning-based wood defect detection has garnered substantial attention from both academia and industry. In their research, Shi et al. [15] devised a wood veneer defect detection model employing a multichannel masked regional convolutional neural network (R-CNN). The model comprises a glance network and a multichannel masked R-CNN. To enhance its performance, intermediate features extracted from the glance network are fused using a genetic algorithm. Subsequently, the multi-channel mask R-CNN is utilized for defect classification and localization. The model achieved an impressive overall classification accuracy of 98.70%, with an average classification accuracy of 95.31%. However, one limitation is the extensive training time required, and the model's complexity is also a concern. A wood defect detection algorithm based on the improved YOLOv5 architecture has been proposed by Han et al. [16]. In their work, enhancements were introduced into the YOLOv5 model, including the incorporation of a Transformer Encoder block, Coordinate Attention module, Swin Transformer, and BiFPN structure. These modifications substantially enhanced the model's capability to extract features and fuse multi-scale features. Experimental results demonstrated that the improved model achieved an mAP score of 84.2, representing a 3.1% improvement over the original YOLOv5 model. However, it is worth noting that the model still exhibits suboptimal performance in detecting cracks

and knots with cracks. In order to enhance the feature pyramid network of the You Only Look Once (YOLO) algorithm, Cui et al. [17] incorporated the spatial pyramid pool (SPP) mechanism. They proposed an improved YOLOv3-based network model and further curated and expanded the open-source dataset of wood defects from the University of Oulu. The overall detection accuracy on the expanded dataset achieved an impressive 93.23%, although the accuracy of the model for detecting wood crack defects was slightly lower. Gao et al. [18] introduced a novel classification framework known as Multiscale Feature Fusion-Based Bilinear Fine-Grained Convolutional Neural Network (BLNN), which leverages a feature fine-grained fusion strategy to construct a bilinear classification model. The proposed approach integrates two sub-networks for feature extraction and employs a bilinear join operation to capture intricate image characteristics at a fine-grained level. The model achieves a detection accuracy of 99.20% on an expanded dataset from the University of Oulu. However, experiments were conducted only for one type of defect, wood knots, so there is no way to know the accuracy of the model in detecting other types of wood defects, such as cracks, wormholes, etc.

In summary, among various methods for wood surface defect detection, the target detection algorithm utilizing deep learning emerges as a highly cost-effective approach. It requires only a computer and a high-definition camera to achieve more accurate identification and location of defects. To accurately detect wood surface defects while meeting the application requirements of factories, the YOLOv7 [19] was selected as the overall network model framework in this paper. The YOLO family of algorithms [20–25] represents a well-established single-stage target detection algorithm family, known for its simplicity and efficiency, making it widely adopted across various target detection applications. YOLOv7 is one of the newest and best-performing detection models in the YOLO family of algorithms. It is designed with a trainable bag-of-freebies, which allows YOLOv7 to greatly improve the detection accuracy without increasing the inference cost, while the method of extend and compound scaling is proposed and applied to the YOLOv7, which greatly reduces the number of parameters in the network and improves the detection speed. Before the introduction of the YOLO series, the dominant approach in target detection was the Region-based Convolutional Neural Networks (R-CNN) series algorithm, e.g., RCNN, Fast R-CNN and Faster R-CNN. Despite its high detection accuracy, the R-CNN series received criticism for its slower detection speed, mainly due to its two-stage network structure. However, with the advent of the YOLOv1, a single-stage target detection algorithm, the need for generating candidate frames was eliminated, and the network directly extracted features to predict object classification and location. This breakthrough significantly improved the detection speed of the model while ensuring a certain level of detection accuracy, capturing the attention of numerous scholars and researchers. As a result, YOLO became a popular choice for target detection tasks in diverse fields. However, the YOLO series algorithm has always had the problem of poor detection capability for small targets [26]. For wood surface defects, the defects are usually small, and there is a large variability in the shape and size of the same defects, as well as a large similarity between different types of defects. This brings a great challenge to the accurate detection of wood defects by the YOLO algorithm. To enhance the ability of the YOLO algorithm to capture small defects, a novel attention mechanism named Omni-Dynamic convolution Coordinate Attention (ODCA) is proposed in this paper. Simultaneously, an enhancement is introduced to improve the precision of defect detection. This improvement involves the development of an innovative features extraction network block called S-HorBlock. By integrating S-HorBlock into both the Backbone and Head components of the YOLO network, the model gains a more refined ability to differentiate between various types of defects. As a result, the defect detection accuracy of the YOLO algorithm is significantly enhanced, aligning it with the factory's stringent requirements for wood defect detection accuracy.

The paper is structured into several sections, each focusing on different aspects of the research. In Section I, the importance of the study is emphasized, highlighting the contributions made in this work. It also presents a comprehensive review of existing wood

defect detection techniques, with a particular emphasis on the in-depth analysis of deep learning-based methods for surface defect detection on wood. Moving on to Section II, a thorough exposition of the proposed ODCA attention mechanism is provided, along with the details of the modified model based on YOLOv7, specifically designed for the accurate detection of surface defects in wood. In Section III, we present the results obtained from comparison and ablation experiments, shedding light on the model's performance. Finally, Section IV presents the conclusions drawn from the research, summarizing the key findings and potential implications.

We present the following contributions in our work:

1. Introducing ODCA, a novel attention mechanism that enhances the network's capability to detect small targets, thus improving feature representation within the network.
2. An omni-dimensional dynamic convolution coordinate attention-based YOLO model (ODCA-YOLO) for wood defects detection is proposed.
3. Designing an efficient features extraction network block (S-HorBlock) specifically for ODCA-YOLO. S-HorBlock enhances the network's learning capacity and improves its ability to extract diverse types of defective wood features.

## 2. Methodology

### 2.1. Omni-Dimensional Dynamic Coordinate Attention

2.1.1. Review of Omni-Dimensional Dynamic Convolution

The convolution layers of traditional CNN tend to learn the data with a single static convolution kernel, but this inevitably results in the "equal treatment" of important and unimportant regions of the data by the convolution kernel. Consequently, the network's sensitivity to crucial image regions decreases, and important information necessary for accurate detection is often overlooked, resulting in a "bluntness" of the network. Moreover, the conventional method to enhance detection accuracy in convolutional neural networks involves increasing network depth or width. However, this approach also escalates the network's computational complexity, making it large, cumbersome, and unsuitable for mobile deployment in factory settings.

The structure of dynamic convolution [27] is as in Figure 1. The weight values of different regions in the image are first learned through a Squeeze-and-Excitation (SE) attention mechanism [28], which is weighted into each of the $n$ convolution kernels and combined linearly for them, and then these weighted $n$ convolution kernels are learned. This results in different convolution weights for different inputs, thus improving the accuracy of the CNN and performing effective inference. Although the dynamic convolution increases the learning of $n$ convolution kernels, this only increases the complexity of the network without increasing the depth and width of the network. Dynamic convolution is the use of parallel $n$ convolution kernels sharing output channels through aggregation, and only needs to compute the attention $\{\pi_k(x)\}$ and aggregation kernels, which is negligible compared to convolution. The dynamic convolution finds a better compromise between network performance and computational load and improves the expressiveness of the model by fusing multiple convolution kernels and aggregating them in a nonlinear manner via attention. In Figure 1, "+" denotes a linear combination of $n$ convolution kernels, Softmax stands for the Softmax activation function, GAP stands for global average pooling, FC stands for fully connected, and ReLU stands for the ReLU activation function, Here, "*"denotes convolution operation, $W_i$ stands for the $i$-th convolution kernel, and $\alpha_{wi}$ represents the attention scalar associated with the convolution kernel $W_i$.

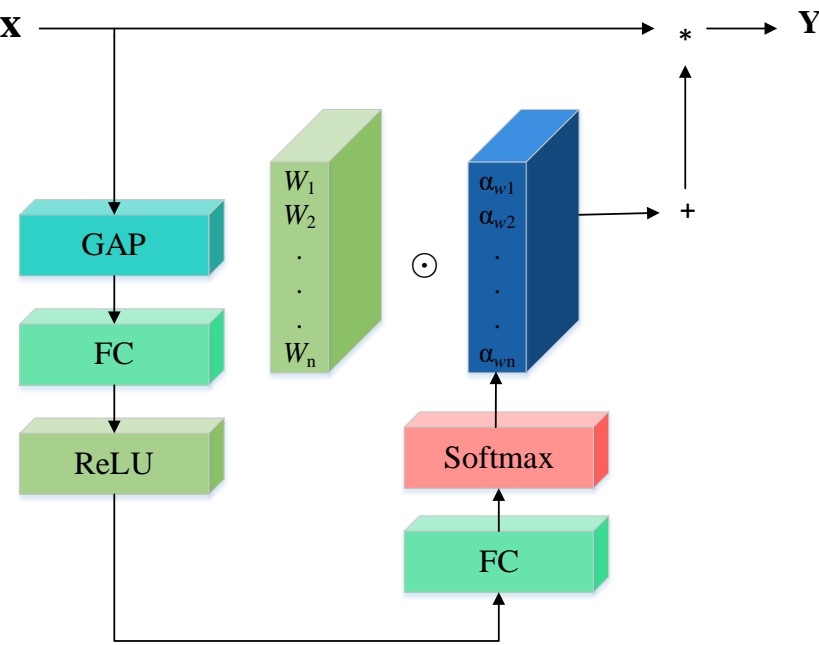

**Figure 1.** Structure of dynamic convolution.

Figure 2 illustrates the structure of Omni-Dimensional dynamic convolution (OD-Conv) [29]. Building upon dynamic convolution, we introduce three additional dimensions: the spatial dimension of the convolution kernel, the input channel dimension, and the output channel dimension. These dimensions are also endowed with dynamic characteristics. This augmentation not only boosts the feature extraction capacity of the convolution layer but also significantly improves the network's ability to capture rich contextual cues in comparison to dynamic convolution.

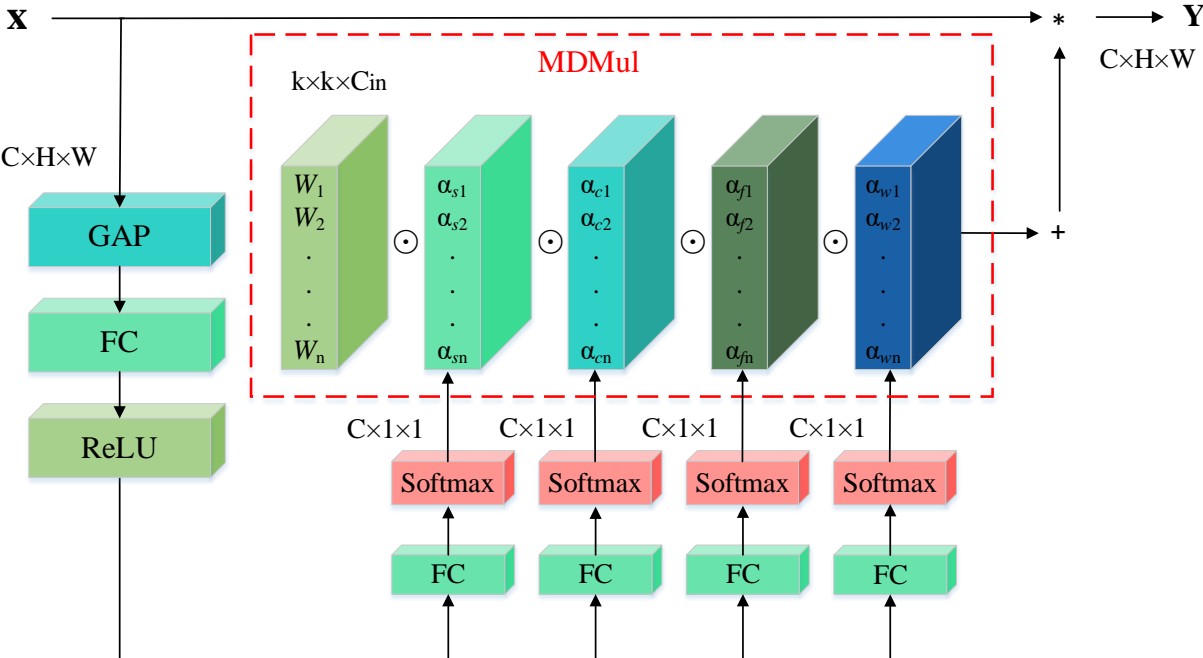

**Figure 2.** Structure of ODConv.

The application of ODConv for defect detection can be represented by the following procedure:

The feature map of the defects can be described as:

$$\mathbf{X} = \text{Input } (\mathbf{X} \in \mathbb{R}^{H \times W \times C}), \tag{1}$$

The above feature map is input into SE, and a mapping operation is performed, that is:

$$\mathbf{U} = \text{F}_{\text{tr}}(\mathbf{X}) \ (\mathbf{X} \in \mathbb{R}^{H \times W \times C}), \tag{2}$$

where $\mathbf{U}$ is the output, and can be written as $\mathbf{U} = [\mathbf{u}_1, \mathbf{u}_2, \cdots, \mathbf{u}_C]$. $\text{F}_{\text{tr}}(\ )$ is a primary mapping and can be taken as a convolution operator.

Thus we have,

$$\mathbf{u}_i = \mathbf{v}_i * \mathbf{X} = \Sigma_{s=1}^{C'} \mathbf{v}_i^s * \mathbf{x}^s, \tag{3}$$

where $\mathbf{u}_i \in \mathbb{R}^{H \times W} (i = 1, 2, \cdots, C)$ is the $i$-th output, $\mathbf{V} = [\mathbf{v}_1, \mathbf{v}_2, \cdots, \mathbf{v}_C]$ to denote the learning set of the filter kernel. $\mathbf{v}_i$ $(i = 1, 2, \cdots, C)$ refers to the parameters of the $i$-th filter, and $\mathbf{v}_i = [\mathbf{v}_i^1, \mathbf{v}_i^2, \cdots, \mathbf{v}_i^{C'}]$. $\mathbf{X} = [\mathbf{x}^1, \mathbf{x}^1, \ldots, \mathbf{x}^{C'}]$ is the feature map of the defects. $\mathbf{v}_i^s$ $(s = 1, 2, \cdots, C')$ is a 2D spatial kernel representing a single channel of $\mathbf{v}_i$ acting on the corresponding channel of $\mathbf{X}$. Here, * denotes convolution operation.

The feature vector $z$ is obtained by compressing the corresponding spatial information on each channel into the corresponding channel using the global average pooling in the SE structure, that is,

$$z_c = \mathbf{F}_{sq}(\mathbf{u}_c) = \frac{1}{H \times W} \sum_{i=1}^{H} \sum_{j=1}^{W} u_c(i, j), \tag{4}$$

where $z_c$ is the $c$-th element of $z$, and $(z \in \mathbb{R}^{H \times W})$.

The output $z$ is obtained by subjecting its output to two fully connected layers and two activation function layers, so we have,

$$s = \sigma(\text{L}_2 \, \delta(\text{L}_2 z)), \tag{5}$$

In the proposed model, we introduce the weight matrix $\mathbf{s}$, which represents the weight ratio of each channel. $\text{L}_1$ and $\text{L}_2$ refer to the two fully connected layers, while $\delta$ and $\sigma$ denote the ReLU and sigmoid activation functions, respectively.

In dynamic convolution, the attention weight $\mathbf{s}$ is defined as $\pi_k$, which represents the attention weight of the $k$-th convolution kernel. Unlike before, the attention weight is now applied not to individual channels but to the entire convolution kernel.

In contrast, ODConv extends this concept by applying attention weights to the entire convolution kernel and its corresponding three dimensions. This allows for a more comprehensive weight assignment, enhancing the model's capability to capture relevant information from the convolution layers.

In the ODConv, the s can be defined as $\pi_i(x)$, and

$$\pi_i(x) = \text{s} \ (0 \leqslant \pi_i(x) \leqslant 1, \ \sum_{n=1}^{n} \pi_i(x) = 1), \tag{6}$$

Then, the attention weights $\pi_i$ are input to the Multi-dimensional multiplication operation (MDMul) module (as shown in Figure 2) for the convolution kernel of ODConv together with $\mathbf{X}$ for convolution operations in each dimension. Given $n$ convolution kernels, the corresponding kernel space has four dimensions regarding the spatial kernel size k × k, the input channel number $c_{\text{in}}$ and the output channel number $c_{\text{out}}$ for each convolution kernel, and the convolution kernel number $n$.

Subsequently, the attention weights $\pi_i$ are combined with $\mathbf{X}$, forming the input for the Multi-dimensional multiplication operation (MDMul) module (as depicted in Figure 2). This operation takes place for the convolution kernels of ODConv, incorporating convolution operations across each dimension.

With $n$ convolution kernels, the kernel space now encompasses four dimensions: the spatial kernel size k × k, the input channel number $c_{\text{in}}$, the output channel number $c_{\text{out}}$ for each convolution kernel, and the total number of convolution kernels $n$. This comprehensive representation characterizes ODConv. The ODConv can be described as:

$$\mathbf{Y} = \left(\alpha_{w1} \odot \alpha_{f1} \odot \alpha_{c1} \odot \alpha_{s1} \odot W_1 + \cdots + \alpha_{wn} \odot \alpha_{fn} \odot \alpha_{cn} \odot \alpha_{sn} \odot W_n\right) * \mathbf{X}, \tag{7}$$

In this context, $\alpha_{wi} \in \mathbb{R}$ represents the attention scalar associated with the convolution kernel $W_i$. The ODConv model builds upon dynamic convolution by introducing three novel attention scalars, computed along different dimensions within the kernel space of $W_i$. These dimensions include the spatial dimension $\alpha_{si} \in \mathbb{R}^{c_{in}}$, the input channel dimension $\alpha_{ci} \in \mathbb{R}^{c_{in}}$, and the output channel dimension $\alpha_{fi} \in \mathbb{R}^{c_{out}}$. The symbol $\odot$ denotes multiplication operations performed across these various dimensions. To calculate $\alpha_{wi}$, $\alpha_{fi}$, $\alpha_{ci}$, and $\alpha_{si}$, we employ the multi-head attention module $\pi_i(x)$ as a crucial component of the process.

The pseudo-code for the ODConv process is shown in Algorithm 1.

---

**Algorithm 1:** ODConv

---

**Input:** $\mathbf{X} \in \mathbb{R}^{H \times W \times C}$
**Output:** $\mathbf{Y} \in \mathbb{R}^{H \times W \times C}$
# Initialization
Step 1: $\mathbf{x} \leftarrow$ Input
Step 2: $\mathbf{u}_i \leftarrow \mathbf{v}_i * \mathbf{X} \leftarrow \Sigma_{s=1}^{C'} \mathbf{v}_i^s * \mathbf{x}^s$
Step 3: $z_c \leftarrow \mathbf{F}_{sq}(\mathbf{u}_c) \leftarrow \sum_{i=1}^{H} \sum_{j=1}^{W} u_c(i,j)/H \times W$
Step 4: $\mathbf{z} \leftarrow [z_1, z_2, \ldots, z_c]$
Step 5: $\mathbf{s} \leftarrow \sigma(L_2\delta(L_2z)), \delta \leftarrow Relu, \sigma \leftarrow sigmoid$
Step 6: $\pi_i(x) \leftarrow \mathbf{s}$
Step 7: $\alpha_{w1} \leftarrow \pi_i(x), \alpha_{f1} \leftarrow \pi_i(x), \alpha_{c1} \leftarrow \pi_i(x), \alpha_{s1} \leftarrow \pi_i(x), \alpha_{wn} \leftarrow \pi_i(x), \alpha_{fn} \leftarrow \pi_i(x), \alpha_{cn} \leftarrow \pi_i(x), \alpha_{sn} \leftarrow \pi_i(x)$
Step 8: $\mathbf{Y} \leftarrow (\alpha_{w1} \odot \alpha_{f1} \odot \alpha_{c1} \odot \alpha_{s1} \odot W_1 + \ldots + \alpha_{wn} \odot \alpha_{fn} \odot \alpha_{cn} \odot \alpha_{sn} \odot W_n) * \mathbf{X}$

---

### 2.1.2. Design of ODCA

We note that both the dynamic convolution and the ODConv introduce dynamic convolution operations, which are based on the Squeeze-and-Excitation (SE) attention mechanism. The SE attention mechanism mainly squeezes all information into a channel of feature vectors through global average pooling and then performs two fully connected operations and two nonlinear activations to obtain the normalized weight values corresponding to each channel. For the dynamic convolution and the ODConv, these weight values need to be used to perform the corresponding convolution calculation together with the convolution kernel. Nevertheless, the utilization of the SE attention mechanism unavoidably results in the loss of location information during the squeezing phase. Consequently, when applying dynamic convolution and ODConv to determine the attention weight values for distinct regions of the image, location information is compromised, leading to an incomplete utilization of input data in subsequent convolution operations.

In contrast to the SE attention mechanism, Coordinate Attention (CA) [30], shown in Figure 3, is a novel attention mechanism tailored for lightweight networks. It uniquely incorporates location information into the channel attention, enabling the networks to efficiently attend to a larger area, thereby enhancing accuracy without significant additional computational burden. The proposed CA mechanism decomposes channel attention into two parallel one-dimensional feature encoding processes. This enables the effective integration of spatial coordinate information into the resulting attention maps. Specifically, CA utilizes two 1D global pooling operations to aggregate input features along the vertical and horizontal directions, generating two separate direction-aware feature maps. These maps contain direction-specific details and are further encoded as two attention maps. Each attention map captures long-range dependencies of the input feature maps along a specific spatial direction, thereby preserving the essential location information. Multiplying the attention maps by the input feature map enhances its representation. This distinct approach, which distinguishes spatial directions and generates coordinate-aware feature maps, is referred to as CA.

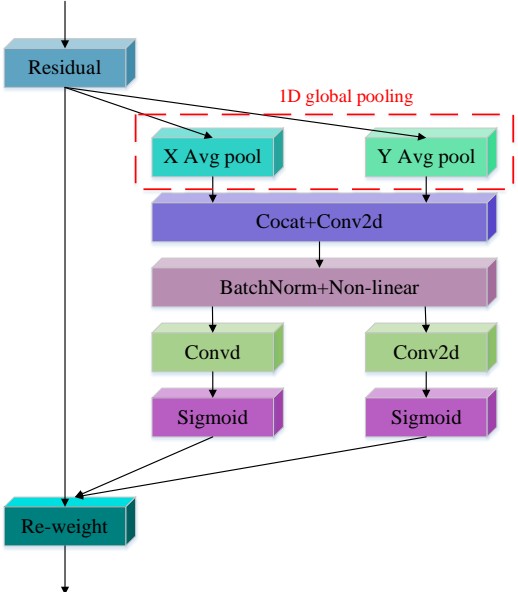

**Figure 3.** Coordinate Attention (CA) structure.

Unlike previous attention methods utilized in lightweight networks, Coordinate Attention (CA) offers distinct advantages. Firstly, it excels at capturing not only cross-channel information but also direction-aware and location-aware details. This enriched information enhances the model's ability to accurately locate and identify the target of interest. Secondly, the CA boasts flexibility and lightweight attributes, making it seamlessly adaptable for integration into classical modules.

In view of the above analysis, it is necessary to solve the problem of location information loss caused by the squeezing operation when dynamic convolution and full-dimensional dynamic convolution are used with the SE attention mechanism. Also inspired by Vision Transformer's encoding of location and based on CA, we embed the MDMul module of ODConv into the CA, and a novel attention mechanism, named Omni-Dimensional dynamic Coordinate Attention (ODCA), is constructed. The structure of ODCA is shown in Figure 4.

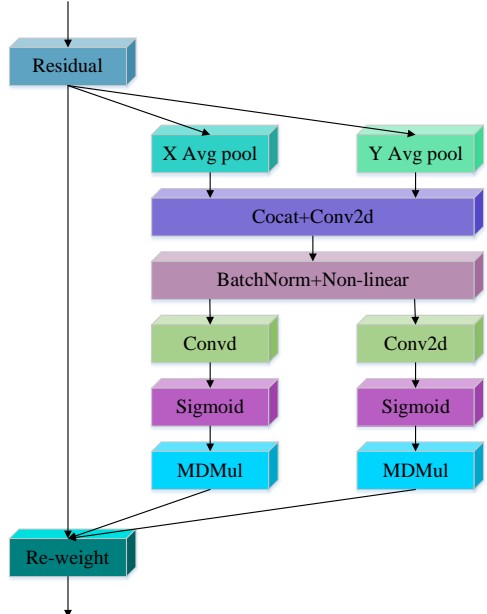

**Figure 4.** ODCA module structure.

The ODCA is a dynamic attention module that seamlessly combines ODConv with CA, effectively integrating both *x*-direction and *y*-direction components to obtain separate attention weights. This approach strategically retains the location information within the feature map, enabling dynamic convolution to leverage the input's location information. Consequently, the dynamic convolution becomes more sensitive to the detection target, enhancing the model's ability to precisely locate the target of interest and improving the model's expressiveness.

The application of the ODCA for defect detection can be outlined as follows.

Given the input **X**, the global average pooling operation is performed for each of the *x*-direction and *y*-direction of the feature map to obtain two feature vectors of $C \times H \times 1$ and $C \times 1 \times W$, respectively. We have,

Given the input **X**, we perform a global average pooling operation for both the *x*-direction and *y*-direction of the feature map. This yields two distinct feature vectors with dimensions of $C \times H \times 1$ and $C \times 1 \times W$, respectively. We have,

$$z_c^h(h) = \frac{1}{w} \sum_{0 \le i < W} x_c(h, i), \tag{8}$$

$$z_c^w(w) = \frac{1}{h} \sum_{0 \le i < H} x_c(j, w), \tag{9}$$

Among these, $z_c^h$, $z_c^w$ represents the output of the *c*-th channel at the height h and the output of the *c*-th channel at the width w, respectively.

Then a splicing and 1×1 convolution operation is performed on them, followed by a BatchNormal process and nonlinear activation, yielding

$$\mathbf{f} = \delta(F_1[\mathbf{z}^h, \mathbf{z}^w]), \; (\mathbf{f} \in \mathbb{R}^{C/r \times (H+W)}), \tag{10}$$

where **f** is the intermediate feature map that encodes spatial information in both the *x*-direction and the *y*-direction. [●, ●] denotes the concatenation operation along the spatial dimension. $\delta$ is the nonlinear activation function. $F_1$ is the $1 \times 1$ convolution operation. *r* is the hyperparameter representing the reduction ratio for controlling the block size as in the SE block, which is used to reduce the number of channels to reduce the model complexity.

In the proposed approach, we utilize the intermediate feature map **f** to encode spatial information in both the *x*-direction and the *y*-direction. To achieve this, we employ the concatenation operation [●, ●] along the spatial dimension, followed by the nonlinear activation function $\delta$. Subsequently, $F_1$, representing the $1 \times 1$ convolution operation, is applied. The hyperparameter *r* controls the block size, similar to the SE block, and is used to reduce the number of channels, thereby enhancing model simplicity.

Next, a separation operation is executed on the intermediate feature map to derive feature vectors for both directions, specifically for the *x*-direction and the *y*-direction.

$$[\mathbf{f}^h, \mathbf{f}^w] = \mathbf{f}, \; (\mathbf{f}^h \in \mathbb{R}^{C/r \times H}, \mathbf{f}^w \in \mathbb{R}^{C/r \times W}), \tag{11}$$

Subsequently, we convolve the two feature vectors with a $1 \times 1$ convolution kernel, followed by normalization using a sigmoid activation function. This process yields two distinct attention vectors for the *x*-direction and the *y*-direction, respectively.

$$\mathbf{g}^h = \sigma(F_h(\mathbf{f}^h)), \mathbf{g}^w = \sigma(F_w(\mathbf{f}^w)), \tag{12}$$

In this context, we define $\mathbf{g}^h$ and $\mathbf{g}^w$ as $\pi_i(x)$ and $\pi_i(y)$, respectively. These values represent the attention weights for the *x*-direction and *y*-direction, as well as for the *i*-th convolution kernel and its corresponding four dimensions. In other words, $\mathbf{g}^h$ and $\mathbf{g}^w$ signify the attention weights allocated to each dimension within the convolution kernel for both the *x*-direction and *y*-direction, respectively.

$$\pi_i(x) = \mathbf{g}^h, \pi_i(y) = \mathbf{g}^w, \tag{13}$$

The obtained attention weights $\pi_i(x)$, $\pi_i(y)$ are input to the MDMul module of ODConv together with **X** to perform the convolution operation in each dimension, and the output **Y** is

$$
\begin{aligned}
\mathbf{Y} = \; &(\alpha_{hw1} \odot \alpha_{hf1} \odot \alpha_{hc1} \odot \alpha_{hs1} \odot W_1 + \ldots + \alpha_{hwn} \odot \alpha_{hfn} \odot \alpha_{hcn} \odot \alpha_{hsn} \odot W_n) * \mathbf{X} \\
&\times (\alpha_{ww1} \odot \alpha_{wf1} \odot \alpha_{wc1} \odot \alpha_{ws1} \odot W_1 + \ldots + \alpha_{wwn} \odot \alpha_{wfn} \odot \alpha_{wcn} \odot \alpha_{wsn} \odot W_\mathrm{n}) * \mathbf{X}
\end{aligned} \tag{14}
$$

where $\odot$ denotes multiplication operations in different dimensions along the kernel space, and $\times$ represents the corresponding position multiplication. $\alpha_{hwi}$ ($i = 1, 2, \cdots, n$) denotes the attention scalars assigned to the entire convolution kernel in the horizontal attention direction. $\alpha_{hfi}$ denotes different attention scalars assigned to the $c_{\mathrm{out}}$ channel of each convolution kernel. $\alpha_{hci}$ denotes different attention scalars assigned to the $c_{\mathrm{in}}$ channel of each convolution kernel. $\alpha_{hsi}$ denotes different attention scalars assigned to the spatial location of each k $\times$ k convolution kernel. These are obtained from $\pi_i(x)$ in the horizontal attention direction. $\alpha_{wwi}$ denotes the attention scalars assigned to the entire convolution kernel in the vertical attention channel. $\alpha_{wfi}$ denotes different attention scalars assigned to the $c_{\mathrm{out}}$ channel of each convolution kernel. $\alpha_{wci}$ denotes different attention scalars assigned to the $c_{\mathrm{in}}$ channel of each convolution kernel. $\alpha_{wsi}$ denotes different attention scalars assigned to the spatial location of each k $\times$ k convolution kernel. These are obtained from $\pi_i(y)$ in the vertical attention direction.

The pseudo-code for the ODCA process is shown in Algorithm 2.

---

**Algorithm 2: ODCA**

---

**Input:** $\mathbf{X} \in \mathbb{R}^{H \times W \times C}$
**Output:** $\mathbf{Y} \in \mathbb{R}^{H \times W \times C}$
# Initialization
Step 1: $\mathbf{X} \leftarrow$ Input
Step 2: $z_c^h(h) \leftarrow \frac{1}{W} \Sigma_{0 \le i < W} x_c(h, i), z_c^w(w) \leftarrow \frac{1}{h} \Sigma_{0 \le i < H} x_c(j, w)$
Step 3: $\mathbf{f} \leftarrow \delta(F_1[\mathbf{z}^h, \mathbf{z}^w])$
Step 4: $[\mathbf{f}^h, \mathbf{f}^w] \leftarrow \mathbf{f}$
Step 5: $\mathbf{g}^h \leftarrow \sigma(F_h(\mathbf{f}^h)), \mathbf{g}^w \leftarrow \sigma(F_w(\mathbf{f}^w))$
Step 6: $\pi_i(\mathrm{x}) \leftarrow \mathbf{g}^h, \pi_i(\mathrm{y}) \leftarrow \mathbf{g}^w$
Step 7: $\alpha_{hw1} \leftarrow \pi_i(x), \alpha_{hf1} \leftarrow \pi_i(x), \alpha_{hc1} \leftarrow \pi_i(x), \alpha_{hs1} \leftarrow \pi_i(x), \alpha_{hwn} \leftarrow \pi_i(\alpha x), \alpha_{hfn} \leftarrow \pi_i(x), \alpha_{hcn} \leftarrow \pi_i(x), \alpha_{hsn} \leftarrow \pi_i(x)$
$\alpha_{ww1} \leftarrow \pi_i(x), \alpha_{wf1} \leftarrow \pi_i(x), \alpha_{wc1} \leftarrow \pi_i(x), \alpha_{ws1} \leftarrow \pi_i(x), \alpha_{wwn} \leftarrow \pi_i(x), \alpha_{wfn} \leftarrow \pi_i(x), \alpha_{wcn} \leftarrow \pi_i(x), \alpha_{wsn} \leftarrow \pi_i(x)$

Step 8: $\mathbf{Y} \leftarrow (\alpha_{hw1} \odot \alpha_{hf1} \odot \alpha_{hc1} \odot \alpha_{hs1} \odot W_1 + \ldots + \alpha_{hwn} \odot \alpha_{hfn} \odot \alpha_{hcn} \odot \alpha_{hsn} \odot W_n) * \mathbf{X}$
$\quad\quad \times (\alpha_{ww1} \odot \alpha_{wf1} \odot \alpha_{wc1} \odot \alpha_{ws1} \odot W_1 + \ldots + \alpha_{wwn} \odot \alpha_{wfn} \odot \alpha_{wcn} \odot \alpha_{wsn} \odot W_\mathrm{n}) * \mathbf{X}$

---

### 2.2. S-HorBlock Module

To enhance the detection accuracy of our proposed ODCA-YOLO model, we introduced the S-HorBlock structure (depicted in Figure 5a). This novel block combines the HorBlock module from the HorNet network with the ShuffleNetv2 network [31]. The integration of these components aims to further optimize the model's performance in accurately detecting wood defects.

The HorNet network was first proposed by Rao et al. [32]. Based on the concept of high-order spatial interaction, they proposed the recursive gated convolution named **g$^\mathbf{n}$Conv** convolution. The structure of the **g$^\mathbf{n}$Conv** convolution is shown in Figure 5c. The **g$^\mathbf{n}$Conv** convolution achieves the spatial interaction of any order, which can improve the modeling ability and high-density prediction performance of the network model. Accordingly, following the same architecture as Vit [33] and SwinTransformer [32], the HorNet network [34] is built. The HorNet network (as shown in Figure 5b) contains a spatial hybrid layer HorBlock and a feedforward network (FFN). Nonetheless, the HorBlock structure is intricate, and its repeated utilization slows down the model's inference speed. To strike a balance between accuracy and inference speed, we incorporated the ShuffleNetv2 network structure as the overarching framework for S-HorBlock. Within the proposed ODCA-YOLO model, we applied the S-HorBlock module at the initial and final stages of the backbone section, replacing the Efficient Layer Aggregation Networks (ELAN) module

in the head section with the S-HorBlock module. This strategic adaptation ensures both accuracy and improved inference speed for the model.

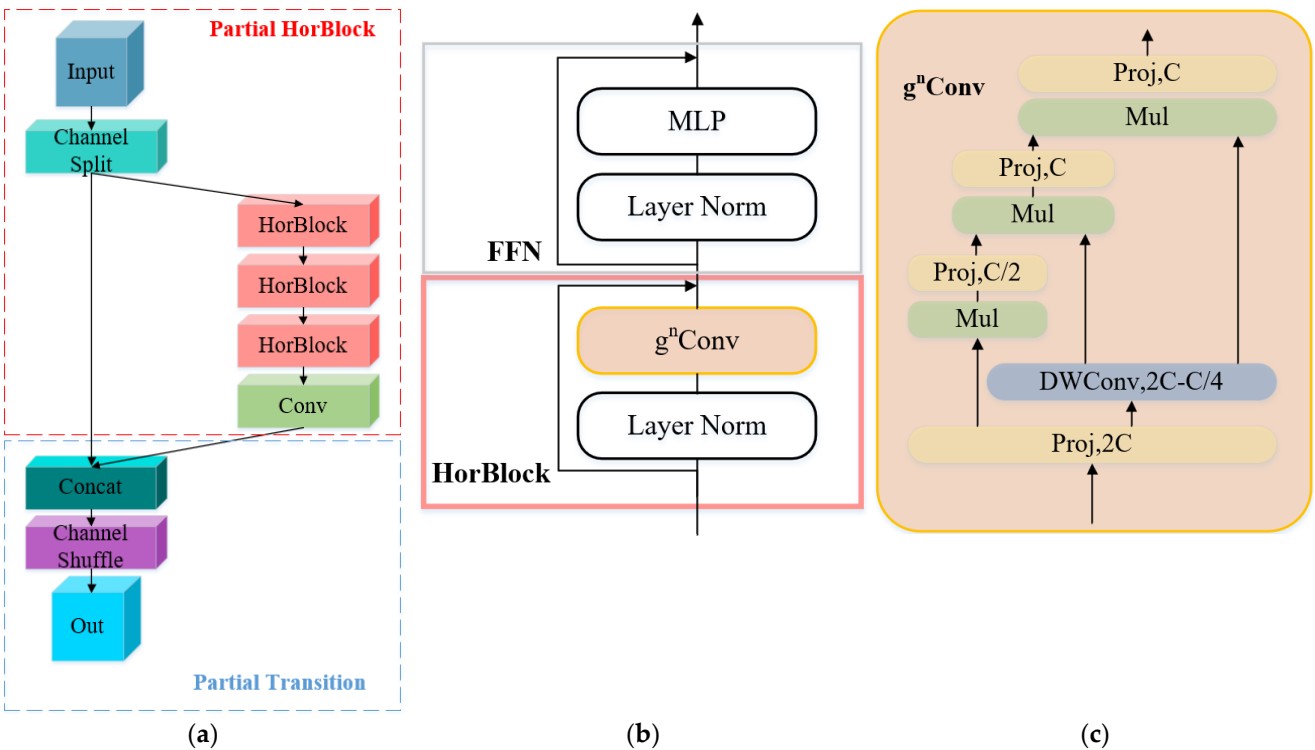

**Figure 5.** Structure of S-HorBlock: (**a**) S-HorBlock module; (**b**) HorNet network structure; (**c**) $g^n$Conv module structure.

### 2.3. The Proposed ODCA-YOLO

In order to be able to detect wood surface defects accurately, using YOLOV7 as the detection framework, an Omni-dynamic convolution coordinate attention-based YOLO (ODCA-YOLO) model is proposed, as shown in Figure 6. In ODCA-YOLO, we aim to precisely locate small defects on the wood surface. To achieve this, we incorporate ODCA in the head section. However, to enhance the overall detection accuracy of the network, we also utilize the S-HorBlock module in both the backbone and head sections, partially replacing the role of the ELAN module. This strategic integration enables the model to better extract wood surface defect features.

CBS denotes the combination of three layers, Convolution Layer, Batch Normalization Layer, and ReLU Activation Function, which is used to extract features. The MP module is used for downsampling. The ELAN module is an efficient network structure designed by YOLOv7 for extracting features. Upsample is an upsampling module. The SPPCSPC module is used to increase the sensory field of the network, which enables the network to adapt to images with different resolutions. The Concat layer is used for the merging of the number of channels. RepConv is a reparameterization structure that speeds up the inference of the network. ImpConv is used to learn the implicit knowledge in the network. The detection module is used to localize and classify the detected targets. The S-HorBlock module is the module designed in this paper that is used to extract and fuse features.

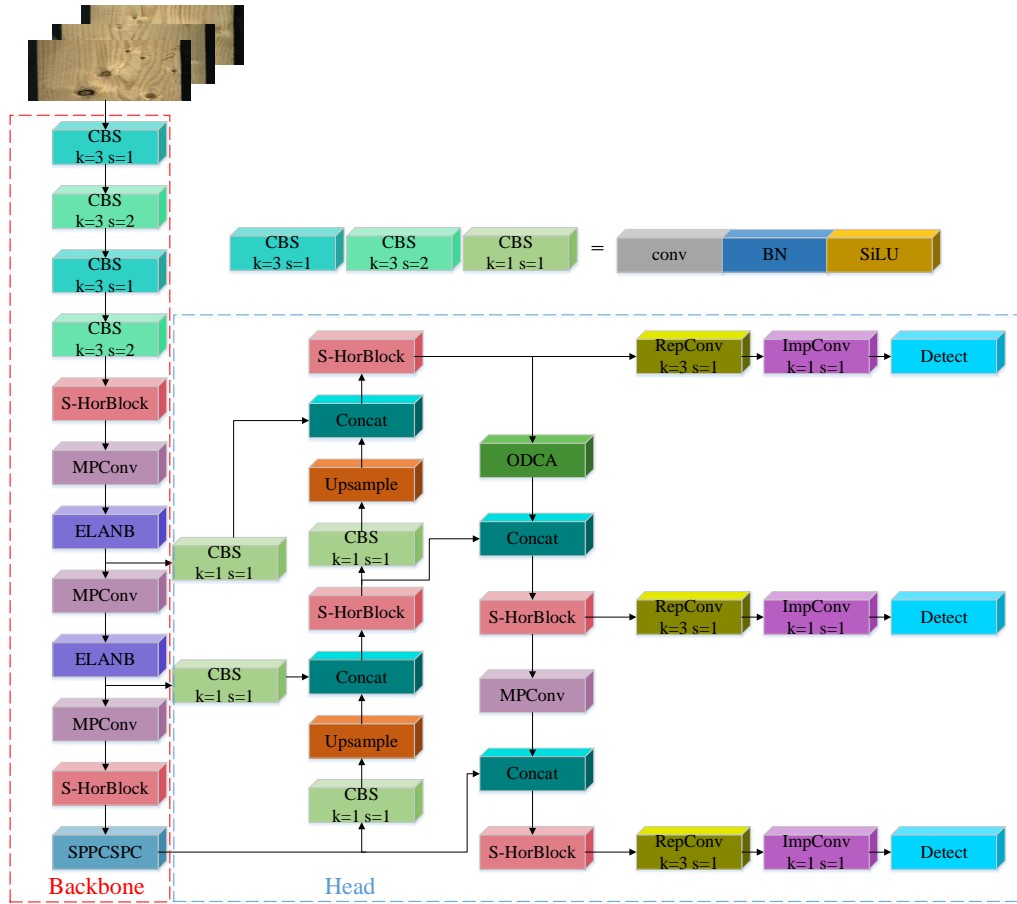

**Figure 6.** Structure of the proposed ODCA-YOLO.

## 3. Experiment and Results

In this section, we provide a comprehensive description of the experimental setup, including details about the experimental environment, the dataset used, and the evaluation metrics employed. Subsequently, we present and analyze the experimental results to validate the effectiveness of our proposed ODCA-YOLO model.

### 3.1. Experimental Details and Dataset

We conducted training and testing of our proposed model using the PyTorch 1.13.0 deep learning framework, utilizing the accelerated computing architecture CUDA 11.6. The experimental setup included a Windows 11 operating system, an Intel(R) Core(TM) i5-9300 CPU with a clock speed of 2.40 GHz (It is manufactured by Intel Corporation, headquartered in Santa Clara, California.), and an NVIDIA GeForce GTX 1650 graphics card (It is manufactured by NVIDIA Corporation, headquartered in Santa Clara, California.). For the deployment environment, we utilized Python 3.10, set the batch size to 4, and conducted training for 200 epochs.

For the experiments, we utilized a large-scale image dataset of wood surface defects, which was generated by VSB-Technical University of Ostrava specifically for automated visual quality control processes [35]. This original dataset contains a total of 20,275 images, consisting of 1992 images without any defects and 18,283 images with one or more surface defects. The dataset encompasses ten types of wood defects, namely Live_Knot, Dead_Knot, Quartzity, Knot with crack, Knot missing, Crack, Overgrown, Resin, Marrow, and Blue stain. We screened 3600 images from the original dataset to construct a new dataset of wood surface defects, excluding the three defects that rarely appear, which are Quartzity, Blue stain, and Overgrown. The image size was 2800 × 1024. The wood surface defect dataset was split into training and test sets in a 9:1 ratio, with an additional 10% of the

training subset used as the validation set. For detailed information on the wood defect distribution in the constructed dataset, please refer to Table 1.

**Table 1.** Distribution of defects in the dataset.

| Defect Type | Number of Occurrences | Number of Images with the Defect | Images with the Defect in the Dataset (%) |
|---|---|---|---|
| Live_Knot | 4070 | 2256 | 62.7 |
| Marrow | 206 | 191 | 5.3 |
| Resin | 650 | 523 | 14.5 |
| Dead_Knot | 2934 | 1875 | 52.1 |
| Knot_with_crack | 542 | 398 | 11.1 |
| Knot_missing | 121 | 110 | 3.1 |
| Crack | 517 | 371 | 10.3 |
| without any defects | — | 7 | 0.2 |

*3.2. Performance Evaluation*

To evaluate the accuracy of wood defect detection, we utilized two metrics: average precision (*AP*) and mean average accuracy (*mAP*). These performance evaluation criteria allow us to assess the effectiveness of our approach. The calculation methods for these metrics are explained in detail below.

The *AP* metric evaluates the performance of our proposed model for each specific wood defect category. It is computed as follows:

$$Precision = \frac{TP}{TP + FP},$$ (15)

$$Recall = \frac{TP}{TP + FN},$$ (16)

$$AP = \int_0^1 P(R)dR,$$ (17)

In Equations (15) and (16), *TP* stands for true-positive defects, *FP* represents false-positive defects, and *FN* indicates false-negative defects.

The *mAP* represents the average accuracy of recognition across all wood defect categories and is calculated as follows:

$$mAP = \frac{\sum_{i=1}^{c} AP_i}{c},$$ (18)

In this experiment, c denotes the number of wood defect categories, which is seven, and *i* represents a specific defect category. The *AP* measures the average accuracy of recognizing a single category, specifically *AP*@0.5, which indicates the area under the precision-recall (P–R) curve for that particular category. The *mAP*@0.5 is obtained by summing the average recognition accuracy, *AP*@0.5, across all categories and calculating the average value.

*3.3. Ablation Experiments*

To assess the practical significance of the proposed enhancements in wood defect detection, we conducted four sets of ablation experiments on a curated dataset, employing consistent environmental and parameter settings. These joint ablation experiments were performed to evaluate the effectiveness of the modifications. The results of these experiments are presented in Table 2, where we used the YOLOv7 model as the reference benchmark for comparison against the last three experimental sets. The YOLOv7+S-HorBlock represents that the S-HorBlock module is used at the beginning and end of the Backbone section of the original YOLOv7, and the S-HorBlock module is used in the

Head section instead of the role of the ELAN module. The YOLOv7+ODCA represents the ODCA-based YOLOv7 model without the S-HorBlock module, and Ours represents the proposed algorithm.

**Table 2.** Ablation results.

| | mAP | AP | | | | | | |
| | | Live_Knot | Morrow | Resin | Dead_Knot | Knot_with_Crack | Knot_Missing | Crack |
|---|---|---|---|---|---|---|---|---|
| YOLOv7 | 0.694 | 0.777 | 0.811 | 0.669 | 0.789 | 0.486 | 0.632 | 0.693 |
| YOLOv7+S-HorBlock | 0.745 | 0.830 | 0.747 | 0.698 | 0.832 | 0.543 | 0.868 | 0.694 |
| YOLOv7+ODCA | 0.753 | 0.842 | 0.807 | 0.793 | 0.836 | 0.592 | 0.736 | 0.668 |
| ODCA-YOLO | 0.785 | 0.835 | 0.930 | 0.790 | 0.834 | 0.614 | 0.782 | 0.707 |

Note: AP = Average precision; mAP = Mean AP; YOLO= You Only Look Once.

From the ablation experiments, we can see that the introduction of the S-HorBlock module and ODCA module demonstrates a relatively large improvement in the accuracy of the whole algorithm to identify wood defects. By incorporating the S-HorBlock module, the mAP value improves by 5.1%. Similarly, the introduction of the ODCA module leads to a 5.9% increase in the mAP value. When both modules are utilized in the model, the mAP value reaches 78.5%, showing a remarkable 9.1% improvement compared to the original model. The proposed ODCA-YOLO model exhibits significant enhancements in detecting the seven wood defects, surpassing the performance of the original model.

### 3.4. Comparisons with Other Methods and Experiments

Figure 7 displays the Precision-Recall (P–R) curves of the YOLOv5, the original YOLOv7, and the ODCA-YOLO models for detecting the seven types of defects in the constructed wood defect dataset. The P–R curve is a crucial performance indicator, with a larger enclosed area indicating better performance. Remarkably, as depicted in Figure 7, our proposed ODCA-YOLO model outperforms the other two models, showcasing superior defect detection performance.

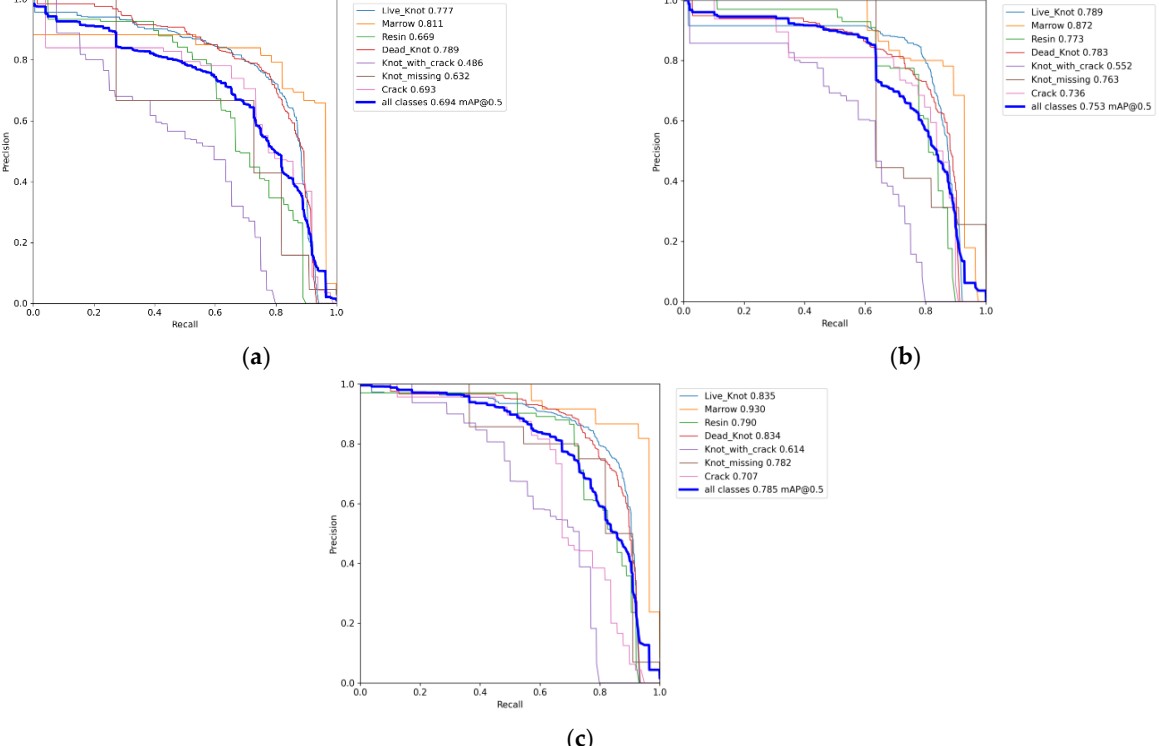

**Figure 7.** Precision–recall (P–R) curves: (**a**) YOLOv7; (**b**) YOLOv5; (**c**) ODCA-YOLO.

Figure 8 showcases the comparative visualization results of the detection outcomes among the mentioned models. The superiority of the ODCA-YOLO model in accurately locating and detecting all wood defects is evident, as it yields more precise predicted bounding boxes. Furthermore, in contrast to the other two models, the ODCA-YOLO model significantly reduces false detections and missed detections, leading to a notable improvement in overall detection performance.

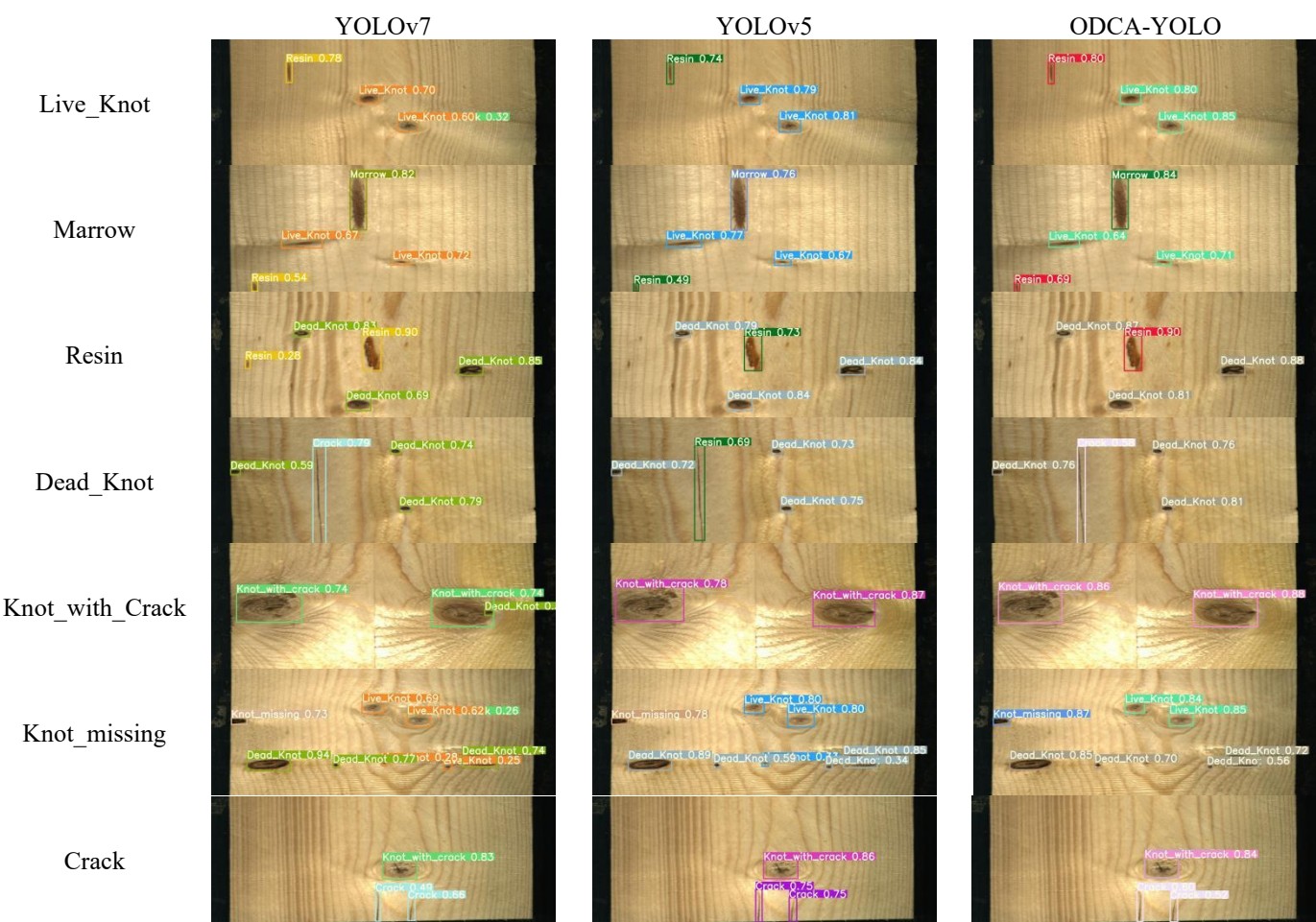

**Figure 8.** Comparison of detection results.

Figure 9 presents the comparative analysis of the Grad-CAM results obtained from the aforementioned models. Grad-CAM is a technique used to generate class activation heat maps, depicting the contribution distribution of input images to the output prediction. In these heat maps, locations with a more pronounced and deeper red color indicate a higher response and greater contribution to the network at corresponding positions in the input images. By observing Figure 9, it is evident that the ODCA-YOLO model exhibits superior accuracy in localizing the regions associated with wood defects.

To further validate the superior detection accuracy of the proposed ODCA-YOLO algorithm, we conducted a comparison experiment with five other mainstream defect algorithms, namely SSD, YOLOv5, YOLOX, YOLOv7, and RetinaNet. The experimental results are presented in Table 3.

The comparative experimental findings further validate that the improved model surpasses other conventional algorithms in terms of accuracy for wood defect detection.

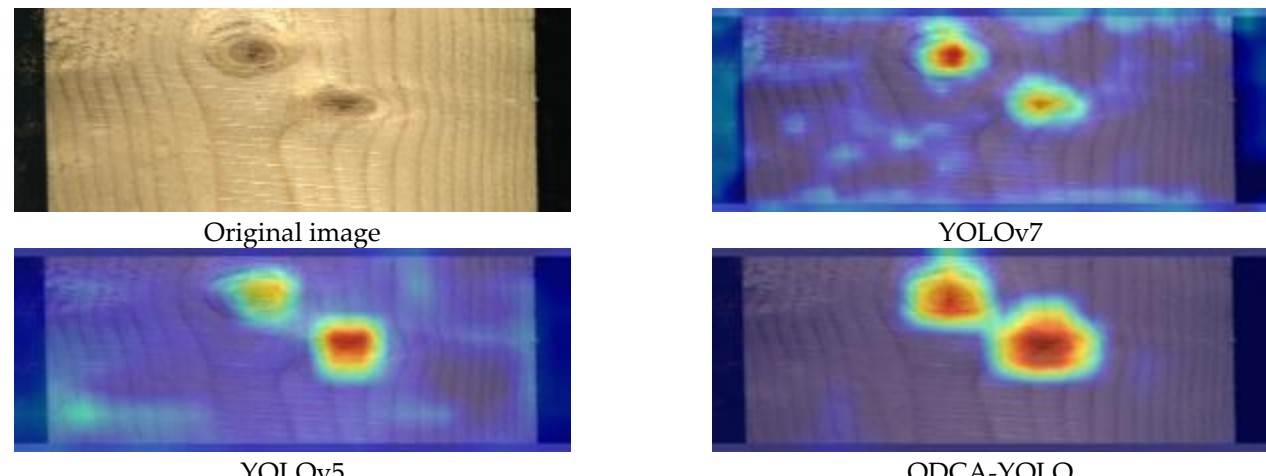

**Figure 9.** Gradient-weighted Class Activation Map (Grad-CAM).

**Table 3.** Comparison results.

| | mAP | | | | AP | | | |
|---|---|---|---|---|---|---|---|---|
| | | Live_Knot | Morrow | Resin | Dead_Knot | Knot_with_Crack | Knot_Missing | Crack |
| YOLOv5 | 0.753 | 0.789 | 0.872 | 0.773 | 0.783 | 0.552 | 0.763 | 0.736 |
| YOLOv7 | 0.694 | 0.777 | 0.811 | 0.669 | 0.789 | 0.486 | 0.632 | 0.693 |
| YOLOX | 0.600 | 0.692 | 0.661 | 0.760 | 0.666 | 0.403 | 0.474 | 0.544 |
| SSD | 0.605 | 0.695 | 0.642 | 0.774 | 0.650 | 0.511 | 0.483 | 0.479 |
| RetinaNet | 0.526 | 0.684 | 0.413 | 0.735 | 0.633 | 0.541 | 0.477 | 0.196 |
| ODCA-YOLO | 0.785 | 0.835 | 0.930 | 0.790 | 0.834 | 0.614 | 0.782 | 0.707 |

## 4. Conclusions

In this study, the large-scale dataset of the VSB-Technical University of Ostrava was filtered and optimized to obtain a more streamlined and suitable dataset for application and to provide data support for research in the field of wood defect detection. To enhance YOLOv7's ability to detect small target defects, we integrate CA attention with ODConv, resulting in the dynamic attention mechanism ODCA. Additionally, we introduce the S-HorBlock structure, inspired by ShuffleNet, to replace the ELAN structure in YOLOv7, further improving the algorithm's recognition accuracy. Experimental results on the optimized wood defect dataset demonstrate significant improvements, with ODCA-YOLO achieving a mAP value of 78.5%, 9.1% higher than the original YOLOv7 algorithm. Compared to single-stage target detection algorithms SSD and RetinaNet, ODCA-YOLO outperforms them with an 18% and 25.9% increase in mAP values, respectively. For YOLOv5, except for Crack defects, ODCA-YOLO shows noticeable improvements in detecting the other six types of defects. Moreover, compared to the YOLOX algorithm, ODCA-YOLO demonstrates enhanced detection accuracy for all seven types of defects. In summary, although our method performs well for the detection of surface defects in wood, it has limitations in visualizing the internal condition of the wood, so our future research will attempt to integrate deep learning methods into techniques that can characterize the internal wood condition.

**Author Contributions:** R.W.: Supervision, Writing—review & editing, Funding acquisition, Investigation, Methodology. F.L.: Writing—original draft, Software, Methodology. B.W.: Supervision, Conceptualization, Validation, Data curation. X.M.: Resources, Writing—review, Funding acquisition. All authors have read and agreed to the published version of the manuscript.

**Funding:** This study was co-supported by the Science and Technology Planning Project of Guangxi Province, China (No. 2022AC21012); the industry–university–research innovation fund projects of China University in 2021 (No. 2021ITA10018); the fund project of Key Laboratory of AI and Information Processing (Hechi University), Education Department of Guangxi Zhuang Autonomous Region (No. 2022GXZDSY101); the Natural Science Foundation Project of Guangxi, China (No. 2018GXNSFAA050026).

**Data Availability Statement:** Data will be made available on request.

**Conflicts of Interest:** The authors declare no conflict of interest.

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
