# Peer review of "ODCA-YOLO: An Omni-Dynamic Convolution Coordinate Attention-Based YOLO for Wood Defect Detection"

_forests, doi:10.3390/f14091885_

Round 1

Reviewer 1 Report

The article shows an interesting topic of wood defect detection based on computer vision.  The proposed YOLO model contains some innovations and its improved performance is well compared with other models.  However, the quality of presentation is needed to be improved, as the figures are not precisely described.  Other comments are given for a major revision:

1. In the abstract, for The results clearly demonstrate the effectiveness of our approach, showing an impressive 9% improvement in mean average precision (mAP) compared to the original algorithm. Our proposed model achieves a comprehensive performance in wood surface defect detection., it would be better to mention the exact mAP achieved by the proposed model; it is necessary to specify the comprehensive performance like accuracy, recall, and speed.

2. For the discussions of current methods of wood defect detection, a few latest articles from 2020 can be mentioned:

(1)    Brunela Pollastrelli Rodrigues, Christopher Adam Senalik, Xi Wu, James Wacker, Use of Ground Penetrating Radar in the Evaluation of Wood Structures: A Review, Forests 2021, 12(4), 492.

(2)    Qiwen Qiu, Thermal conductivity assessment of wood using micro computed tomography based finite element analysis (μCT-based FEA), NDT & E International 2021, 139, 102921.

Please include some discussions of the latest development of characterization techniques in the literature review.

3. The proposed method is suitable for detecting the wood surface defects, but it has limitation in visualizing the internal condition of wood.  This indicates that the defect detection is not comprehensive enough.  So, the reviewer suggests that the future work would be the incorporation of the deep learning method into the techniques capable of characterizing the internal wood condition.

4. For “As a consequence, it has gradually been eliminated.”, this statement is not convincing because manual visual recognition is still an important way for today’s initial check of wood products.

5. For “It is not possible to know the accuracy of the model for other types of defects such as cracks.”, why is it not possible to get the accuracy? Crack detection by YOLO-based methods has widely been applied in these years.

6. On page 3, the fast R-CNN and faster R-CNN should also be mentioned after the discussion of R-CNN.  Besides, explain the selection of YOLOv7 instead of other versions of YOLO.  In other words, some features of YOLOv7 can be mentioned.

7. References about applications of YOLO should be added to support this statement “However, the YOLO series algorithm has always had the problem of poor detection capability for small targets.”.  How to define an image object which is small?

8. Network features of ODCA and S-HorBlock are suggested to be mentioned in the introduction section.  Why are they so unique for this work?

9. More details and explanations can be included in Fig. 1.  For example: “GAP”, “FC”, “*” and “+”.

10. The spatial dimension of the convolution kernel, the input channel dimension, and the output channel dimension can be mentioned in Fig. 2.

11. Quality in Fig. 5 is poor, as the text is unclear.

12.  In section 2.1.2. Design of ODCA, please indicate “two parallel one-dimensional feature encoding processes”, “1D global pooling operations” in Fig. 3.

13. Figure 2 is repeated?

14. Some contents in Fig. 6 may be missing.

15. For “As a result, the ODCA-YOLO model excels at effectively extracting and identifying wood surface defects.”, this is problematic.  No result is shown to confirm this statement in this section.

16. In Table 2, labelling of proposed model is needed, instead of using “ours”.

17. The results in Fig. 8 are not clear for reading.

18. In Table 3, it would be better to add errors/deviations for the model’s performance.

Moderate editing of English language required.

Author Response

Dear Editors and Reviewers:

Thanks for your letter and for reviewer’s comments concern our manuscript entitled “ODCA-YOLO:An Omni-dynamic convolution coordinate attention based YOLO for wood defect detection” (forests-2587635). Those comments are valuable and helpful for revising and improving our paper, as well as the important guiding significance to our researches. We have studied all comments carefully and have made correction which we hope meet with approval. Revised portion are marked with underline in the paper. The main corrections in the paper and the responds to the reviewers’ comments are In the following files.

Reviewer 2 Report

1. Figure 1: + operation should get at least two inputs, in Fig. 1 + node gets one input only! Moreover, GAP symbol is not defined.

2. ODConv component is described by the authors in three ways:

- visual (Figure 2 at line 198),

- algorithmic (the pseudocode after line 260), and

- mathematical notation (text in lines 200-260).

However, the notation, the symbols, operations, etc. used in all those forms are not mutually consistent. It creates the chaotic impression, and cannot be left in this form in the paper reporting state of the art results on object detectors in images.

3. Line 298: Figure 2 --> Figure 4 (change of figure number). However, this figure is the same as Figure 3. This mess must be corrected.

4. Figure 5, its part (c) is different than the original Rao [30] paper for gConv module. It has inconsistent channel values.

5. Figure 5: change raster to vector format

6. Figure 6 is the main contribution of the submission. It presents modified YOLO v7 architecture for object detection by insertions of S-HorBlocks in many places of its design. Despite the use of component names from YOLO, readers will appreciate brief explanations of component names used there.

7. Table 1: (a) what you mean by over occurrence, (b) make headings shorter.

8. Table 2 and 3: headings too long.

9. Figure 7: most legend entries are cut off and therefore they are not readable. However, moving the external location of each legend into the left lower corners of the figure parts, will give enough place not to cut off the legend entries.

10. Figure 8: must be rearranged as now we get illegible defects descriptions.

1. "bluntness" of the network ??? --> "rigidity" (as antonym of "fexibility")

2. "elevate" precision???

3. Line 190, caption to Figure 1: convolutional --> convolution

Author Response

Dear Editors and Reviewers:

Thanks for your letter and for reviewer’s comments concern our manuscript entitled “ODCA-YOLO:An Omni-dynamic convolution coordinate attention based YOLO for wood defect detection” (forests-2587635). Those comments are valuable and helpful for revising and improving our paper, as well as the important guiding significance to our researches. We have studied all comments carefully and have made correction which we hope meet with approval. Revised portion are marked with underline in the paper. The main corrections in the paper and the responds to the reviewers’ comments in the following files.

Reviewer 3 Report

In the paper, the authors discuss an approach called the Omni-dynamic convolution coordinate attention-based YOLO (ODCA-YOLO) model for the identification of surface defects in wood/timber. The experimental results are presented, confirming the effectiveness of the method.

Comments on the manuscript are listed below:

1. General remark: the manuscript was not prepared with due diligence.

2. Figure 6 is missing.

3. Caption Figure 4 instead of Figure 2 (line 298).

4. The descriptions in the Figures are illegible (too small font, white colour).

5. Explain acronyms when they are first used (e.g. BLNN line 106, SE line 176, ELAN line 390).

6. Line 442 'In this equation' - which one exactly? Above are the equations 15, 16, and 17.

7. Relu or ReLU? - please unify.

8. The subscripts and superscripts in Algorithms (lines 260, 371) are illegible.

9. Table 2 and 3 - compilation of experimental results is inconclusive. In the table headings there is (%). Thus, instead of e.g. 0.785 it should be 78.5.

10. Some formulae (e.g. 1, 15, 16, 17) and the References are not in line with publisher's template.

11. Please, pay attention to the explanations of the symbols in the equations (subscripts/italic e.g. lines 253, 255, 329, 336).

12. Suggestion - 'ODCA-YOLO:' in the title is unnecessary.

Thorough proofreading is necessary. Check punctuation, also in formulas.

Author Response

Dear Editors and Reviewers:

Thanks for your letter and for reviewer’s comments concern our manuscript entitled “ODCA-YOLO:An Omni-dynamic convolution coordinate attention based YOLO for wood defect detection” (forests-2587635). Those comments are valuable and helpful for revising and improving our paper, as well as the important guiding significance to our researches. We have studied all comments carefully and have made correction which we hope meet with approval. Revised portion are marked with underline in the paper. The main corrections in the paper and the responds to the reviewers’ comments are in the flowing flies.

Round 2

Reviewer 1 Report

The manuscript has been revised and improved greatly.  The quality is good and the paper is acceptable at its present form.

Author Response

Dear reviewer:

Thank you very much for taking the time to review this manuscript.

Kind regards.

Rijun WANG 

Reviewer 2 Report

In the revised version authors referred to my all comments to the original version. The current version is more clear.

Author Response

(The authors gave the same response as above.)

Reviewer 3 Report

I read carefully all the authors' answers and I traced the changes in the text.

There is a significant improvement; however, I still think that the manuscript is not prepared with due diligence, e.g., Figure 5 is lost (there are only descriptions left), Tables 2 and 3 are a mess, I do not see any changes made to the equations and references (publisher's template).

Minor editing of the English language is required.

Author Response

Dear reviewer:

Thank you very much for taking the time to review this manuscript.

Kind regards

Rijun WANG